# The Emerging Roles of Extracellular Chaperones in Complement Regulation

**DOI:** 10.3390/cells11233907

**Published:** 2022-12-02

**Authors:** Nicholas J. Geraghty, Sandeep Satapathy, Mark R. Wilson

**Affiliations:** 1Molecular Horizons and School of Chemistry and Molecular Bioscience, University of Wollongong, Northfields Avenue, Wollongong, NSW 2522, Australia; 2Illawarra Health and Medical Research Institute, Northfields Avenue, Wollongong, NSW 2522, Australia; 3Blavatnik Institute of Cell Biology, Harvard Medical School, Boston, MA 02115, USA

**Keywords:** extracellular chaperone, proteostasis, complement system, neurodegeneration, protein folding, protein aggregation

## Abstract

The immune system is essential to protect organisms from internal and external threats. The rapidly acting, non-specific innate immune system includes complement, which initiates an inflammatory cascade and can form pores in the membranes of target cells to induce cell lysis. Regulation of protein homeostasis (proteostasis) is essential for normal cellular and organismal function, and has been implicated in processes controlling immunity and infection. Chaperones are key players in maintaining proteostasis in both the intra- and extracellular environments. Whilst intracellular proteostasis is well-characterised, the role of constitutively secreted extracellular chaperones (ECs) is less well understood. ECs may interact with invading pathogens, and elements of the subsequent immune response, including the complement pathway. Both ECs and complement can influence the progression of neurodegenerative diseases, including Alzheimer’s disease, Parkinson’s disease, Huntington’s disease and amyotrophic lateral sclerosis, as well as other diseases including kidney diseases and diabetes. This review will examine known and recently discovered ECs, and their roles in immunity, with a specific focus on the complement pathway.

## 1. Introduction

Proteostasis (protein homeostasis) includes all those processes that control protein synthesis, folding, localisation and degradation, and plays important roles in organismal function and viability. Molecular chaperones, proteins that assist in protein folding and prevent undesirable protein–protein interactions, are key players in intracellular proteostasis. However, some chaperones are constitutively secreted into the extracellular milieu [1,2,3]. In addition to roles in controlling undesirable protein–protein interactions and the clearance of extracellular misfolded proteins, these extracellular chaperones (ECs) also exert effects on the immune system. In mammals, immunity arises from the operation of two discrete systems, the innate and adaptive immune systems. The former is a rapid, non-specific response which acts by a variety of mechanisms to destroy pathogens and infected cells, and involves mast cells, eosinophils, basophils, neutrophils and macrophages that all differentiate from a common myeloid progenitor. Adaptive immunity is a slower but specific response to pathogens involving T and B cells, which produces circulating antibodies as the primary trigger of pathogen killing. Complement is one of the most important components of the innate immune system and is comprised of a series of circulating zymogens and proteins that when activated assemble on the surface of target cells to form lytic pores. Complement can also be activated by antibodies bound to the surface of bacteria or infected cells, and complement proteins bound to cell surfaces prime (opsonise) those cells for attack by cellular components of the innate immune system. Once activated, innate immune cells can recruit other immune cells to sites of infection by producing cytokines and chemokines, and produce perforin and granzyme, which when released cause lysis of bacteria or infected cells [4]. In addition to the roles of ECs in the clearance of misfolded proteins and immunity, they are also likely to play an important role in complement regulation, and may represent potential therapeutic targets.

## 2. Proteostasis

Following translation, proteins undergo folding to reach their native conformation, which is essential for function [5]. If proteins do not achieve their native conformation, they may expose regions of hydrophobicity to solution, which can lead to unfavourable interactions and the formation of insoluble protein aggregates. Furthermore, proteins can unfold if exposed to unfavourable conditions including elevated temperature [6], extremes of pH [7] and oxidative stress [8]. Misfolding proteins may self-interact to form either unstructured (amorphous) or β-sheet-rich fibrillar (amyloid) aggregates. ATP-independent holdase chaperones bind to regions of exposed hydrophobicity on proteins to inhibit their aggregation, whereas ATP-dependent foldase chaperones actively mediate protein folding and refolding [9]. The processes that operate to maintain proteostasis differ between the intracellular and extracellular compartments of the body.

### 2.1. Intracellular Proteostasis

Inside mammalian cells, 332 genes encode 88 chaperones and 244 co-chaperones, including the “heat shock protein” (HSP) families, the foldases HSP90, 70, 60 and 40, and the holdase small HSP (sHSP), as well as endoplasmic reticulum (ER)- and mitochondrial (mt)-specific chaperones [10]. Despite an abundance of chaperones in the intracellular environment, proteostasis dysfunction or environmental stresses can overload the chaperone network and result in an inability to rescue misfolded proteins, leading to inappropriate and potentially dangerous protein aggregation. Under these conditions, the targeted degradation of misfolded proteins and protein aggregates becomes essential for cell survival. Within cells, this is achieved by two principal systems, the ubiquitin-proteasome system (UPS) and autophagy.

In the UPS, proteins designated for degradation are “ubiquitinated”, a process by which chains of multiple ubiquitin molecules are conjugated to the target protein by ubiquitin ligases [11]; the polyubiquitinated protein can then be shuttled to the cytoplasmic proteasome for degradation. The proteasome consists of a barrel-shaped 20S subunit, containing multiple inwardly-facing proteolytically active sites, and a 26S “lid” which controls the entry of ubiquitinated proteins into the proteolytic chamber [12,13]. Once target proteins have been degraded, the amino acids are recycled by the cell. Autophagy is responsible for the degradation of large protein aggregates and organelles within cells [14]. The cell forms an autophagosome to encompass the aggregate/organelle and fuses this with one or more lysosomes to form an autophagolysosome, within which proteolytic degradation is performed [15] (Figure 1). The preceding is only a very brief outline of some of the most important processes involved in intracellular proteostasis. Detailed reviews of the many interconnected processes that operate together to maintain intracellular proteostasis are provided in [9] and [16].

### 2.2. Extracellular Proteostasis

Despite the abundance of secreted proteins in extracellular compartments of the body, and the fact that serious neurodegenerative and other disease pathologies are associated with inappropriate and excessive aggregation of extracellular proteins [3], the processes involved in maintaining extracellular proteostasis have received far less attention than those operating inside cells [3]. Unlike intracellular proteostasis, where ATP-dependent chaperones can attempt refolding of misfolded proteins, this is much less feasible in extracellular body fluids where the concentrations of ATP are orders of magnitude lower than inside cells [17]. Consequently, the key processes of extracellular proteostasis are independent of ATP and are largely focussed on stabilising extracellular misfolded proteins and directing them for cell uptake and intracellular degradation. Extracellular misfolded proteins and small soluble aggregates are thought to be bound by ECs via their ATP-independent holdase chaperone action [18]. This action neutralises the toxicity of aggregated protein oligomers, and inhibits further protein aggregation [19,20]. Once formed, EC-misfolded protein complexes are cleared from the body by receptor-mediated endocytosis and subsequent lysosomal degradation [17,21,22]. Alternatively, larger extracellular protein aggregates may be degraded by the plasminogen activation system (PAS) (Figure 1) [17,23]. The best known role of the PAS is to cleave the inactive zymogen plasminogen into an active form (plasmin), which subsequently degrades fibrinogen to fibrin [24]. However, the PAS is also activated by a prion protein [25] and amyloid β fibrils, and plasmin can degrade Aβ fibrils [26]. The PAS has also been shown to degrade a variety of misfolded proteins, including antithrombin, α1-protease inhibitor, alpha-2-macroglobulin (α2m), albumin [27], fibrinogen, ovalbumin [28,29], ovotransferrin and superoxide dismutase (SOD1) [23]. It has been shown that two ECs (clusterin and α_2_-macroglobulin) bind to toxic protein fragments released by the plasmin-mediated digestion of amorphous protein aggregates, leading to the proposal that ECs and the PAS may work co-operatively to ensure the timely and safe removal of misfolded and aggregating proteins from body fluids [23].

It is clear that, as is the case inside cells, chaperones are vital cogs in the processes essential to maintain extracellular proteostasis and organismal viability. Although some intracellular chaperones can be released at low levels from cells, their scarcity in extracellular body fluids suggests that they are very unlikely to provide substantial protection against protein aggregation in this location [17]. There are, however, a number of chaperone proteins that are constitutively secreted from cells, and are present in abundance in plasma and other body fluids in the ECs.

## 3. Extracellular Chaperones (ECs) Implicated in Complement Regulation

The first identified and most studied mammalian EC is clusterin (CLU), which is expressed by many cells, has many different binding partners, and plays diverse roles in different body systems [2]. Haptoglobin (HP) and α2-macroglobulin (α2m) are also known as abundant ECs. Other identified ECs include serum amyloid protein (SAP), neuroendocrine protein 7B2 (7B2), proprotein convertase subtilisin (proSAAS), prosurfactant protein C (ProSP-C), integral transmembrane protein 2B (BRI2) and pregnancy zone protein (PZP). Recently, the chaperone activities of two ECs, transthyretin (TTR) and neuroserpin (NS), were reported to be amyloid-selective, meaning that they inhibit amyloid formation but not amorphous protein aggregation. Further putative ECs were recently identified in a study that selectively retrieved from human serum proteins that were bound to a misfolding protein immobilised on magnetic beads. These putative, newly identified ECs include vitronectin (VTN), plasminogen activation inhibitor-3 (PAI-3) and the complement components C1r and C1s [1]. Finally, a recent study of C. elegans identified 57 genes that were up-regulated in response to pathogens, and that when silenced increased extracellular protein aggregation, implicating these genes as being important in extracellular proteostasis in this organism [30]. Subsequently, overexpression of four of these genes was shown to significantly reduce the aggregation of an extracellular protein and to prolong C. elegans lifespan [30]. Although the function of all these C. elegans gene products have yet to be established, they are likely to include ECs. In more complex organisms, including humans, in addition to those already known (Table 1) [2], it is likely that many more ECs are yet to be discovered.

### 3.1. Established ECs

#### 3.1.1. Clusterin (CLU)

Originally named owing to its ability to induce cell clustering in vitro [31], clusterin (CLU, a.k.a Apolipoprotein J, ApoJ) was the first mammalian EC to be identified. Clusterin (CLU) is encoded by the CLU gene, located on chromosome 8 [32], and is a disulfide-linked heterodimeric protein of ~60 kDa [33]. CLU is initially folded in the lumen of the endoplasmic reticulum (ER) before being transported into the Golgi apparatus where it is further glycosylated [34]. In the late Golgi compartment, the CLU polypeptide is cleaved to generate the α- and β-chains [35]. During normal physiological conditions, CLU is secreted from cells [36] and is present in human plasma at ~100–150 μg/mL [37] and cerebrospinal fluid (CSF) at ~2 μg/mL [38]. CLU plays roles in sperm maturation [39], cell survival [40] and differentiation [41], cancer [42] and inflammation [43]. Most notably however, CLU plays an important role in both intracellular and extracellular proteostasis, and is implicated in a number of diseases [44]. CLU inhibits both amorphous and amyloid aggregation of a wide variety of proteins [32,45,46]. The complex structure of CLU is yet to be resolved by conventional techniques [44] and the structural elements responsible for binding to misfolded proteins are still unknown, but may be related to CLU surface hydrophobicity, which is enhanced by decreased pH [47].

CLU can exert neuroprotective effects [48,49,50] and, notably, mutations in the human CLU gene are one of the highest known risk factors for Alzheimer’s disease (AD) [51,52,53]. CLU binds with high affinity to Aβ oligomers to prevent their toxicity [19,20] and aggregation [19,54], and addition of CLU to CSF from AD patients promotes its removal by macrophage-like (U937) cells [55]. In Huntington’s disease, CLU is increased in the serum of patients compared to healthy controls [56], and CLU is also upregulated in the CNS in amyotrophic lateral sclerosis (ALS) [57]. Studies have suggested that CLU is increased in both the serum [58] and CSF [59,60] of Parkinson’s disease patients, but other studies dispute this [61]. Finally, CLU binds to late apoptotic cells (by binding to histones present on the cell surface) and promotes their phagocytosis in vitro and in vivo. CLU KO mice demonstrate susceptibility to apoptotic cell-induced autoimmunity [62]. Serum CLU levels correlate with disease severity in a range of autoimmune diseases including rheumatoid arthritis [63], myocarditis [64], systemic lupus erythematosus [65], diabetes [66] and multiple sclerosis [67]. CLU has also been found to be associated with protein deposits found in other neurological diseases, including in both white matter disease [68] and multiple sclerosis [69].

Serum CLU levels are reduced with the progression of Huntington’s disease, and are inversely correlated with increasing levels of C4 and C7 precursors, as well as C9 [56]. In post-mortem tissue from multiple sclerosis patients, immunohistochemical analysis of multiple sclerosis lesions highlighted increased expression and co-localisation of CLU with complement components C1q, C3b, C4d and MAC, compared to healthy controls [69]. Additionally, in mouse models of cerebral ischaemia, three days after middle cerebral artery (MCA) occlusion, PCR analysis revealed that expression of CLU was increased alongside that of the complement components C1qB and C4 [70]. CLU co-localises with sC5b-9 deposits in the kidneys of glomerulonephritis patients [37]. In a rat model of Heymann nephritis, CLU co-localised with C3 as well as the membrane attack complex (MAC) in the capillary walls of the glomeruli; however, when proteinuria was removed, or MAC deposition was prevented, CLU was not detected [71].

CLU has been shown to inhibit the assembly of MAC when incubated in vitro with erythrocytes bearing C5b-7 and purified C8 and C9 [72]. CLU binds directly to each the terminal complement components C7, C8β and C9b [73] and to circulating immune complexes [74]. The binding of CLU to sC5b-9 was proposed to be essential to maintain the solubility of this complex [74]. CLU was suggested to regulate MAC formation in vivo to prevent its uncontrolled activation [75]. However, a separate study suggests that clusterin may not protect against complement-mediated lysis of cells in vivo [76]. The binding of CLU to bacteria and/or bacterial components may inhibit MAC formation. The binding of CLU to the Streptococcus pyogenes virulence protein SIC [77], the dengue virus non-structural protein 1 [78] and the *Pseudomonas aeruginosa* protein dihydrolipoamide dehydrogenase [79] has been reported to inhibit complement activation and MAC formation, whilst the binding of CLU to *Staphylococci epidermis* J9P was reported to enhance complement-induced bacterial lysis [80]. CLU was also shown to co-localise with MAC in myocardial infarction [81]. Finally, due to the high levels of CLU present in seminal fluid [82], it has been suggested to play a protective role in reproduction preventing complement-mediated lysis in the seminal plasma, on sperm membranes and within the female urogenital tract [83].

#### 3.1.2. Haptoglobin (HP)

Haptoglobin (HP) is transcribed from the *HP* gene on chromosome 16q22 [84], and is expressed as one of three major phenotypes, designated as HP 1-1, HP 2-2 and HP 2-1. HP is synthesised as a single chain and in its simplest form is cleaved to produce two α-chains and two β-chains, covalently linked by disulfide bonds. However, owing to the presence of the Hp1 and Hp2 alleles in humans (which encode two variant α-chains), a number of other structural variants occur [85]. HP is found in most extracellular fluids, including human plasma at 0.3–2.0 mg/mL [86] and CSF at 0.5–2 μg/mL [87]. HP has been shown to inhibit the aggregation of a number of different proteins, which form either amorphous [18,88,89] or amyloid aggregates [90,91]. HP is primarily produced in the liver, and has a primary role in binding to and removing free haemoglobin (Hb) released from red blood cells from the circulation; HP-Hb complexes are cleared via CD163 receptors expressed on macrophages [92]. Structural predictions suggest that HP contains a large hydrophobic region responsible for binding to misfolded proteins [89,93]. HP inhibits the aggregation of Aβ [90] and serum HP levels are reduced in AD patients [91].

In a large proteomic screen of Huntington’s disease patient tissue, the expression of HP was upregulated together with C7 precursor and C9 [56]. Furthermore, in patients, serum HP levels increase during complement activation associated with Salmonella infection, and HP KO mice exhibit a reduced ability to control the infection [94]. The complement system has been suggested to play a role in both diabetes pathogenesis and vascular complications [95]. Relative to diabetic patients with HP2.1 or HP2.2 phenotype, the blood of HP1.1 diabetic patients contains higher levels of pro-inflammatory cytokines and lower levels of anti-inflammatory cytokines. However, the severity of diabetes disease was similar regardless of HP genotpye [96]. In mice, injection of Hb led to C3 and C5b-9 deposition in the kidneys, similar to that observed in hemolytic disease in humans [97], but the deposition of these complement components was prevented by intra-peritoneal injection of HP [97]. Finally, the macrophage complement receptor Mac-1 (CD11b/CD18) binds to complement factor iC3b (cleaved C3b), which acts as an opsonin to stimulate phagocytosis. Intriguingly, HP also binds directly to Mac-1 [98]. The mechanisms by which HP may regulate complement remain to be established.

#### 3.1.3. Alpha-2-Macroglobulin (α2m)

Alpha-2-macroglobulin (α2m) is encoded by the *A2M* gene located on chromosome 12p12.3 [99] and is translated as a 180 kDa polypeptide. Two of these 180 kDa subunits are joined by disulfide bonds to form a dimer, and two disulfide-bonded dimers associate to form the final 720 kDa tetramer [100]. α2m is best known as an inhibitor of a broad range of proteases. Cleavage of the α2m bait region (which contains cleavage sites for many different proteases) [101] results in the formation of a molecular “cage” that sterically entraps proteases and thereby prevents them from accessing their substrates [102]. α2m is found in most extracellular fluids including plasma at 1.5–2 mg/mL [103] and CSF at 1–3.6 μg/mL [104]. The regions and/or structural elements of α2m responsible for its chaperone activity remain unknown. However, the hypochlorite oxidation-induced dissociation of α2m tetramers to dimers enhances the chaperone activity of the molecule, potentially implicating the hydrophobic dimer–dimer interface in the chaperone action [105]. α2m is able to inhibit both the amorphous [106] and amyloid [90,107] aggregation of a variety of client proteins, and can bind Aβ to form α2m-Aβ complexes that are internalised by the LDL receptor-related protein (LRP1) and degraded intracellularly [108,109]. *A2M* polymorphisms have been suggested as a genetic risk factor for Alzheimer’s disease [110,111,112,113,114], although this has been disputed [115,116]. α2m has also been reported to bind to other pathogenic proteins, including prion protein (the causative agent in spongiform encephalopathy) [117] and α-synuclein (which aggregates in the neurons of Parkinson’s disease patients) [118].

Although a primary role of α2m is as a protease inhibitor, α2m produced by immune cells binds to many different cytokines and growth factors [119], and therefore may also play a role in complement regulation. α2m and C3 may share a genetic lineage [120]. In a large proteomic screen of Huntington’s disease patient tissue, similar to HP, α2m expression was increased together with C7 and C9 [56]. Plasma α2m levels increase alongside C1s, C1q, C2 and C3 in a porcine model of ischemic reperfusion injury [121]. In a rat model of acute myocardial infarction (induced by coronary artery ligation), the plasma concentrations of both α2m and MAC increased over time from 4 h post-ligation [122]. *Leptospira* bacteria can evade complement-induced lysis through production of proteases that cleave the complement components C6, C7, C8 and C9. In *Leptospira* cultures, addition of α2m led to inactivation of these proteases, subsequently inhibiting C6-C9 cleavage, promoting complement-mediated removal of *Leptospira* [123]. Additionally, α2m can directly bind *Streptococcus pyogenes* [124,125], *Escherichia coli* [126] and the parasite *Trypanasoma cruzi* [127], but whether α2m plays a role in complement-dependent lysis of these species is yet to be investigated. Chronic lymphocytic leukemia (CLL) patients exhibit IgG-hexamers both in serum and at the surface of B cells, and α2m can directly bind these IgG hexamers, leading to over-activation of the complement system, and promoting CLL pathogenesis [128]. The binding of α2m-IgG hexamer complexes to the cell surface receptor GRP78 [128], a receptor mainly expressed on B cells, and upregulated in CLL patients (compared to healthy controls), occurred only in CLL patients, and lead to chronic activation of the complement system, reducing patients ability to clear malignant cells [129].

In human serum, α2m has been found to bind to mannan-binding lectin (MBL) [130,131] and to form complexes with mannose-binding lectin serine proteases (MASP), both components of the lectin complement pathway. These observations suggest a potential regulatory role for α2m within the lectin pathway [132]. However, although α2m can inhibit MASP-1 and to a lesser extent MASP-2 [133], α2m did not prevent in vitro activation of the lectin pathway (measured as C3 and C4 deposition), suggesting that it may not be a physiological inhibitor of the lectin complement pathway in vivo [134].

### 3.2. Recently Identified Putative ECs

#### 3.2.1. Vitronectin (VTN)

Vitronectin (VTN, also known as S-protein) is a 459-residue acidic glycoprotein [135] of mass 54 kDa, present as either an intact polypeptide, or following proteolytic cleavage, as two polypeptides (one 65 and one 10 kDa) linked by four disulfide bonds [136,137]. VTN consists of an N-terminus containing a somatomedin-B domain, followed by an Arginine-Glycine-Aspartate (RGD) sequence, four haemopexin-like domains and three heparin-binding domains [138]. VTN plays a major role in haemostasis and cell adhesion [139], but has recently been shown to inhibit both the in vitro aggregation of Aβ to form amyloid, as well as the amorphous aggregation of citrate synthase (CS) [1], demonstrating that it also has a chaperone activity similar to CLU. VTN is presently associated with Aβ plaques and neurofibrillary tangles in the brains of Alzheimer’s patients [140]. VTN is significantly upregulated in the cerebral blood vessels of sporadic cerebral amyloid angiopathy patients [54], where it co-localises with Aβ deposits in the arteries [141]. When exposed to the denaturing solvent, 1,1,1,3,3,3-hexafluoro-2-propanol (HFIP), VTN can itself aggregate to form typical amyloid fibrils, and VTN oligomers are toxic to neuronal (SH-SY5Y) cells [141].

VTN is an established regulator of complement activation which binds to C5b-7 to inhibit its insertion into the membrane and the subsequent formation of the MAC [142]. The complex formed when VTN binds to C5b-7 is known as soluble C5b-7 (sC5b-7) and is cleared by the kidneys [143]. VTN can prevent MAC formation by inhibiting C9 polymerisation [142]; however, reports are conflicting as to where and how VTN binds the MAC [144,145]. As a defense against the innate immune system, Dengue fever virus binds VTN to its surface to shield itself from complement attack [146]. VTN is also known to bind to a number of different bacteria. In the cases of *Haemophilus influenzae* [147], *Moraxella catarrhalis* [148] and *Pseudomonas aeruginosa* [149], VTN binding inhibits MAC formation. *Neisseria meningitidis* [150] and *Neisseria gonorrheae* [151] also bind VTN, but whether this is to avoid complement activation has yet to be determined.

#### 3.2.2. Plasminogen Activation Inhibitor-3 (PAI-3)

Plasminogen activator inhibitor 3 (PAI-3), also known as protein C inhibitor (PCI), is expressed as a 57 kDa glycoprotein and belongs to the serpin “superfamily” [152,153]. This family also includes the intracellular chaperone HSP47 and the extracellular chaperone neuroserpin (NS; SERPINI1) discussed below. PAI-3 exhibits a typical serpin structure of three β-sheets and nine α-helices [152] and is found in human serum at 6.3 μg/mL [154] and CSF at 50 ng/mL [154]. PAI-3 primarily acts to inhibit activated protein C, but can also inhibit coagulation by binding to multiple enzymes [155,156,157]. PAI-3 plays a major role in haemostasis, thrombosis, reproduction and, to a lesser extent, immunity, predominantly in tissue repair [158]. A recent study showed that PAI-3 inhibits the in vitro aggregation Aβ and CS to form amyloid and amorphous aggregates, respectively [1]. PAI-3 levels are increased in the brains of AD patients [159], where it is found to be associated with tau neurofibrillary tangles [160]. PAI-3 has also been associated with protein aggregates in multiple sclerosis [161] and Parkinson’s disease [162].

There is currently limited knowledge concerning interactions between PAI-3 and the complement system. PAI-3 has been identified as a novel antimicrobial agent, and direct binding of PAI-3 to the surface of both *E. coli* and *Streptococcus pyogenes* results in bacterial lysis [163]; however, the mechanism of action has not been determined. Furthermore, a recent study showed that in human plasma, PAI-3 is present in extracellular vesicles which co-precipitate with complement components C1q, C3, C3a, C4a and C9 [164]. The mechanisms by which PAI-3 may be involved in regulating complement is a target area for future research.

#### 3.2.3. C1r and C1s

Initiation of the classical complement pathway involves C1q binding to antigen-antibody complexes. Once bound to the pathogen surface, C1q undergoes a conformational change, leading to an association with and activation of C1r. Activated C1r leaves C1s, and the interaction between C1q, C1r and C1s molecules leads to the formation of the C1 complex. Although the primary roles of these molecules are as initiators of the complement pathway, both C1r and C1s have been shown to inhibit the aggregation of Aβ to form amyloid, but not the amorphous aggregation of CS [1]. What role, if any, the chaperone activities of C1r and C1s play in complement regulation remains to be determined. However, it is interesting to note that being positioned at the surfaces of cells subjected to complement attack places these ECs in immediate proximity to what would be a concentrated source of damaged and misfolded proteins released from lysed cells. They are therefore ideally positioned to neutralise toxic protein species released and assist in their safe and efficient clearance from body fluids.

## 4. Possible Mechanisms by Which ECs Influence Complement

This review is the first report to highlight the breadth of interactions between ECs and the complement system. Studies of the mechanism(s) underlying these interactions have yet to be performed. However, we can postulate that a shared inherent property of ECs may be responsible for their interactions with complement components. All ECs are holdase chaperones, meaning that they bind to regions of exposed hydrophobicity on misfolding proteins, to stabilise them in solution and inhibit their progressive aggregation to form dangerous insoluble deposits [9]. The first identified, and one of the most recently identified ECs, CLU and VTN, may provide some clues as to a mechanism underlying EC-complement interactions. As previously mentioned, CLU is known to bind directly to several components of the terminal complement pathway (C7, C8β and C9b) [73], and VTN inhibits C9 polymerisation [142]. All of these terminal complement components are known to expose regions of hydrophobicity to solution during their assembly to form the MAC [165,166]. It is therefore feasible that the binding of CLU and VTN to these regions of exposed hydrophobicity during complement activation is the molecular basis of the mechanism by which they inhibit MAC formation. A similar mechanism may apply to many of the other EC-complement interactions. Although currently speculative, this reasoning at least provides a testable rationale for explaining why so many ECs have been found to be associated with sites of complement activation. An additional reason for this association may relate to an important role for ECs in neutralising toxic misfolded proteins that are likely to be released at sites of complement attack (see Section 3.2.3). Future research is needed to more clearly establish the mechanisms underlying the functional interactions between ECs and complement.

**Table 1 cells-11-03907-t001:** Extracellular chaperones and details of client protein aggregation they are known to inhibit.

Chaperone	Gene	Monomer MW (kDa)	Human UniProt ID	Amyloid-Forming Client(s)	Amorphously Aggregating client(s)
Clusterin (CLU)	CLU	55–60	P10909	Lys [19,45], κ-cas, α-syn, Aβ, ccβω [19], α-lac [32,46],	GST [32,45], Cat [32,45], BSA [32], Ovo, ADH [45], Lys, Calc, β2m [19]
Haptoglobin (HP)	*HP*	86	P00738	Aβ, ccβω, Calc, Lys [90], Cat, γ-crys [88]	CS, GST, Ovo [18]
Alpha-2-macroglobulin (α2m)	*A2M*	180	P01023	Aβ, Lys [90,167] ccβω, calcitonin [90]	No inhibition of CS or CPK [168]
Serum Amyloid Protein (SAP)	*PTX2*	25	P02743	Aβ, β2m [169]	LDH [170]
Neuroendocrine protein 7B2 (7B2)	*SCG5*	27	P05408	IAPP [171], Aβ, α-syn [172]	-
Proprotein convertase subtilisin (proSAAS)	*PCSK1N*	27	Q9UHG2	Aβ [173], α-syn [174]	-
Prosurfactant protein C (ProSP-C)	*SFTPC*	21	P11686	Aβ [175]	-
Integral transmembrane protein 2B (BRI2)	*ITM2B*	14	Q9Y287	Aβ [175], IAPP [176]	-
Pregnancy zone protein (PZP)	*PZP*	360	P20742	Aβ [168]	CS, CPK [168]
Transthyretin (TTR)	*TTR*	55	P02766	Aβ, ccβω, α-syn [177]	No inhibition of CS, CLIC1 or BSA [177]
Neuroserpin (NS)	*SERPINI1*	46	Q99574	ccβω, α-syn [177]	No inhibition of CS, CLIC1 or BSA [177]
Vitronectin (VTN)	*VTN*	75	P04004	Aβ [1]	CS [1]
Plasminogen Activation Inhibitor-3 (PAI-3)	*SERPINA5*	52	G3V4B4	Aβ [1]	CS [1]
Prothrombin (PT)	*F2*	72	P00734	Aβ [1]	No inhibition of CS [1]
Complement component C1r	*C1R*	83	B4DPQ0	Aβ [1]	No inhibition of CS [1]
Complement component C1s	*C1S*	28	P09871	Aβ [1]	No inhibition of CS [1]

Abbreviations: α-lag, α-lactalbumin; α-syn, α-synuclein; β_2_m, β-2 microglobulin; γ-crys, γ-crystallin; κ-cas, κ-casein; Aβ, amyloid β_1-42_; ADH, aldehyde dehydrogenase; BSA, bovine serum albumin; Calc, calcitonin; Cat, catalase; ccβω, coiled–coiled β peptide; Calc, calcitonin; CLIC1, chloride intracellular channel protein; CPK, creatine phosphokinase; CS, citrate synthase; GST, Glutathione-S-transferase; IAPP, islet amyloid polypeptide; Lys, lysozyme; Ovo, ovotransferrin. “-“ denotes not tested.

## 5. Conclusions

The importance of extracellular proteostasis in maintaining the normal function of body systems, including the immune system, is becoming increasingly apparent. CLU, the best studied EC, and VTN (a recently identified putative EC), are well known to interact with multiple components of the complement system. The evidence for effects on complement of the other ECs described is less complete but is emerging and justifies further investigation (Figure 2). ECs play an important role in facilitating the clearance of misfolded proteins, and this type of role may extend to also facilitating pathogen clearance, specifically through activation of the complement system. It is likely that many more ECs are yet be identified and with these discoveries will come enhanced understanding of their roles in organismal defense and homeostasis. A better understanding of the complementary roles of ECs in proteostasis and immunity, particularly regulation of the complement system, has the potential to lead to the development of novel therapeutic strategies to treat conditions such as serious wounds, infections and inflammation. Towards this goal, future research to address the following unanswered questions may prove valuable: (1) What is the molecular mechanism by which some ECs inhibit complement activation and is this a direct consequence of the ability of chaperones to bind to regions of exposed hydrophobicity on complement components? (2) Following complement attack, do ECs assist in the safe clearance and disposal of complement components and proteins released from damaged cells? and (3) Could manipulation of specific EC levels or activities in vivo be used as a therapeutic strategy to treat complement-related pathologies?

## Figures and Tables

**Figure 1 cells-11-03907-f001:**
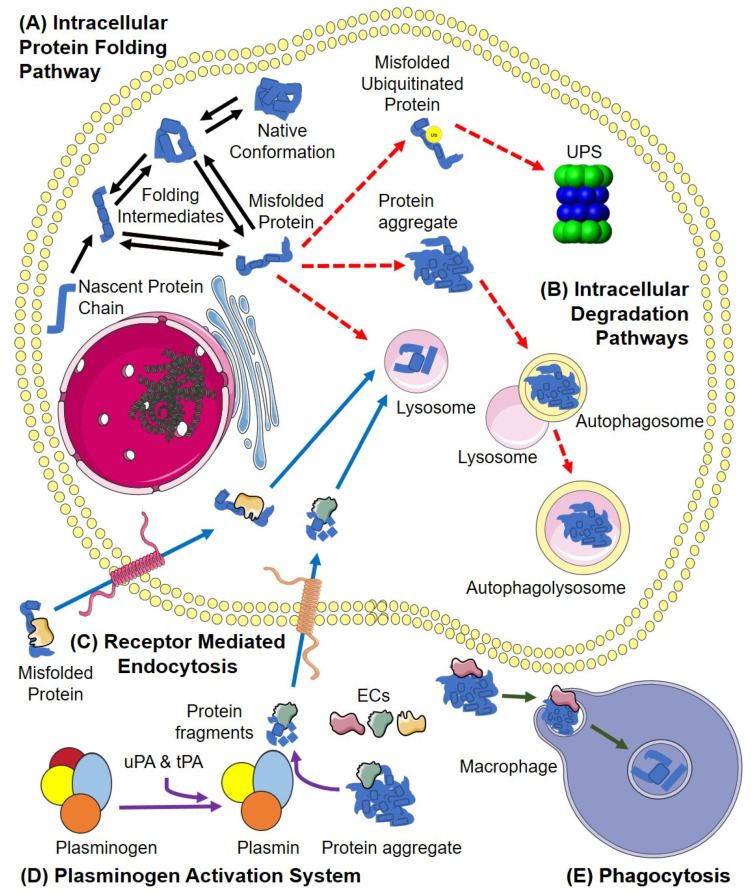
Intra- and Extracellular Proteostasis. (**A**) Intracellularly, proteins need to fold from nascent polypeptide chains into their native conformation (black arrows). If a protein intermediate reaches a misfolded state, and is not rescued by chaperones, it may aggregate (dashed red arrow). (**B**) Large insoluble protein aggregates can be removed via autophagy (digested in autophagolysosomes; dashed red arrow). Alternatively, misfolded proteins may be ubiquitinated and degraded by the ubiquitin-proteasome system (UPS; dashed red arrow). (**C**) Extracellularly, misfolded proteins bound by extracellular chaperones (ECs) are internalised by receptor-mediated endocytosis (RME) and trafficked to lysosomes for degradation (solid blue arrows). Extracellular protein aggregates may be (**D**) digested by the plasminogen activation system (PAS; purple arrows), and the protein fragments generated are bound by ECs also directed for RME and lysosomal degradation (blue arrows) or (**E**) phagocytosed and digested intracellularly (green arrows). Abbreviations: UPS, ubiquitin proteasome system.

**Figure 2 cells-11-03907-f002:**
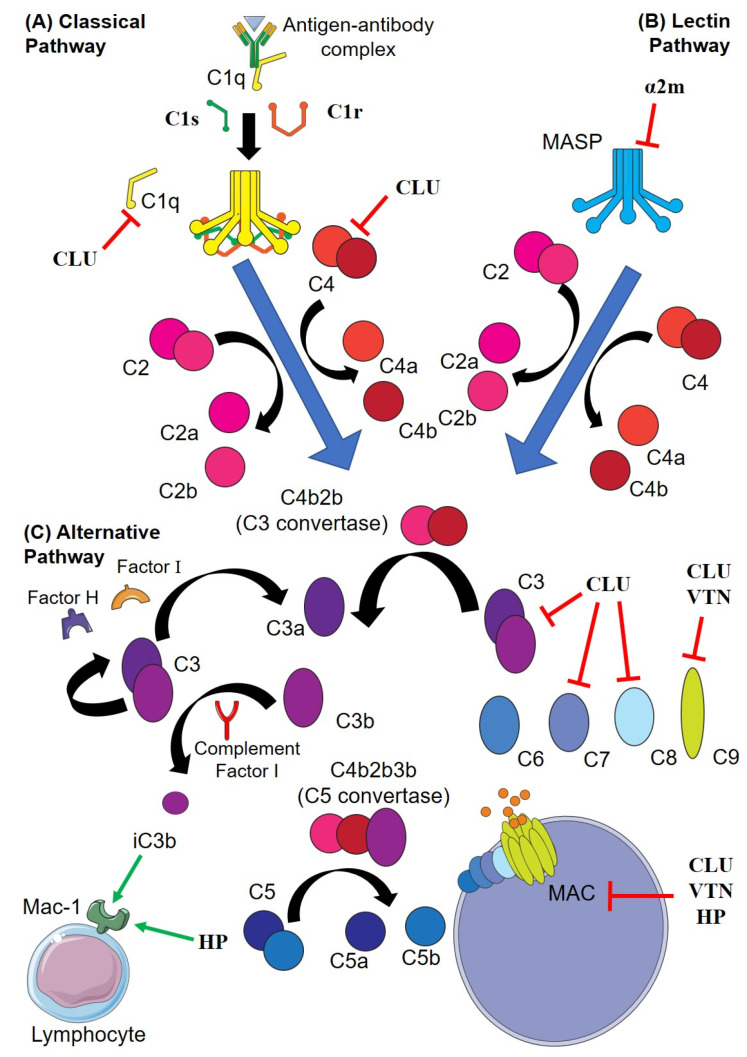
Interactions of extracellular chaperones (EC) and the complement pathway. (**A**) In the classical pathway, C1q undergoes conformational change after binding to IgM or IgG to form antigen-antibody complexes, leading to activation of C1r, which cleaves C1s. (**B**) In the lectin pathway, mannose-binding lectins bind to the pathogen surface and activate the C1r and C1s analogues MASP-1 and -2 (respectively). C1s or MASP-2 subsequently cleave C4 (to C4a and C4b) and C2 (to C2a and C2b); C4b and C2b associate to form the C3 convertase, which binds C3b to form the C5 convertase. The C5 convertase then cleaves C5 to C5a and C5b, C5b recruits C6, C7, C8 and multiple C9 to form the membrane attack complex (MAC). (**C**) In the alternative pathway, C3 spontaneously cleaves to form C3a and C3b, which are sterically inhibited by factor H and factor I, respectively. Complement is activated when C3b binds to a pathogen and cannot be inactivated by factor H. End-capped red lines indicate inhibition of a complement component by an EC (ECs are in bold). Green arrows indicate the direct binding of iC3b and HP to Mac-1, a receptor expressed by a variety of lymphocytes that when bound to iC3b (cleaved C3b) stimulates phagocytosis. Abbreviations; α2m; alpha-2-macroglobulin, CLU; clusterin; HP, haptoglobin; MASP, mannose-binding lectin serine proteases; PAI-3, plasminogen activator inhibitor-3; TTR, transthyretin; VTN, vitronectin.

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
