# Peer review of "The Emerging Roles of Extracellular Chaperones in Complement Regulation"

_cells, 2022, doi:10.3390/cells11233907_

Round 1
Reviewer 1 Report
Geraghty et al. presented a review of recent findings of extracellular chaperones (ECs) and their interactions with the complement system. The study of extracellular chaperones is an emerging field with many unanswered questions. This review may introduce the burgeoning concept to a broad audience in both innate immunology and proteostasis, thus I believe that it should ultimately be published. But the authors should address the following points before publication.
The authors should clarify the potential roles of the ECs in the complement pathways. The current review reads incomplete. It is as if the authors have introduced a group of interesting characters, but before their motives are exposed and before any actions take place, the story ends abruptly.
Specifically, there is a disconnect between the chaperoning functions of the ECs—namely their in vitro ability to prevent aggregation of A and other amyloid-forming peptides—and their physiological role in complement regulation. Is there any evidence that the chaperoning functions of the ECs and their regulatory effect on the complements are connected, or is there a mechanistic hypothesis linking these two? For example, clusterin’s inhibitory effect of amyloid formation may be coincidental and not physiological, even though the same structural features and interaction motifs may be responsible for its inhibitory effect on MAC assembly. Perhaps the ability of ECs to act as holdases account for both observed effects? The authors should clarify this point.
In general, I feel that the review can be strengthened by including more mechanistic discussions that connect the many experimental findings.
Intracellular proteostasis involves a complex network, including both ATP-independent and ATP-driven chaperones. Recent studies suggest that ATP-driven chaperones may use an energy-consuming process to fold and activate proteins into thermodynamically unstable states, thus working as foldases (https://doi.org/10.1038/s41598-018-31641-w, https://doi.org/10.1038/s41589-018-0013-8,https://doi.org/10.1016/j.bpj.2020.08.038, and https://doi.org/10.3390/biom12060832). ATP-independent holdases often pass on their substrates to ATP-driven foldases to complete the folding and activation.
In contrast, the low extracellular ATP concentration (10-100 nM) makes it unlikely to support ATP-dependent chaperones. If ECs only work as holdases, how do they achieve their proposed role in extracellular proteostasis? The authors listed three possible mechanisms: 1)
Plasminogen activation system (PAS) for extracellular degradation, 2) Endocytosis (bringing the misfolded proteins to intracellular lysosome), and 3) Phagocytosis. But it is unclear in the review how ECs induce these activities. The authors should give more details on both the experimental findings and the current mechanistic hypotheses.
The distinction between the intracellular and the extracellular proteostasis should be highlighted. As mentioned above, the intracellular network can “repair” misfolded intracellular proteins by the ATP-driven chaperones, whereas in the current thinking the ECs mostly hold misfolded extracellular proteins for subsequent destruction.
The authors may want to list specific questions regarding the ECs in an expanded Conclusion section to inspire new researchers in this area.
Minor issues:
Line 31: “some chaperones are constitutively secreted into the extracellular milieu.” This needs references.
Line 106-107: “Extracellular misfolded proteins and small soluble aggregates are thought to be bound by ECs and cleared from the body by receptor-mediated endocytosis and subsequent lysosomal degradation.” This needs references.
Line 116-117: “ECs and the PAS may co-operatively work together as key elements of extracellular proteostasis to ensure the timely and safe removal of misfolded and aggregating proteins from body fluids.” Is this purely speculative? Otherwise this needs references.
Author Response
Reviewer 1
Geraghty et al. presented a review of recent findings of extracellular chaperones (ECs) and their interactions with the complement system. The study of extracellular chaperones is an emerging field with many unanswered questions. This review may introduce the burgeoning concept to a broad audience in both innate immunology and proteostasis, thus I believe that it should ultimately be published. But the authors should address the following points before publication.
- The authors should clarify the potential roles of the ECs in the complement pathways.
Specifically, there is a disconnect between the chaperoning functions of the ECs—namely their in vitro ability to prevent aggregation of Ab and other amyloid-forming peptides—and their physiological role in complement regulation. Is there any evidence that the chaperoning functions of the ECs and their regulatory effect on the complements are connected, or is there a mechanistic hypothesis linking these two? For example, clusterin’s inhibitory effect of amyloid formation may be coincidental and not physiological, even though the same structural features and interaction motifs may be responsible for its inhibitory effect on MAC assembly. Perhaps the ability of ECs to act as holdases account for both observed effects? The authors should clarify this point.
We thank the reviewers for the suggestion. An entirely new section (4.0 Possible mechanisms by which ECs influence complement) has been added to the revised manuscript to provide specific discussion of this point.
- If ECs only work as holdases, how do they achieve their proposed role in extracellular proteostasis?
The authors listed three possible mechanisms: 1) Plasminogen activation system (PAS) for extracellular degradation, 2) Endocytosis (bringing the misfolded proteins to intracellular lysosome), and 3) Phagocytosis. But it is unclear in the review how ECs induce these activities. The authors should give more details on both the experimental findings and the current mechanistic hypotheses.
In the revised manuscript (section 2.2 Extracellular Proteostasis), to clarify the mechanisms operating in extracellular proteostasis, we have significantly expanded discussion of the various mechanisms operating and added multiple new citations to provide the reader with relevant supporting publications.
- The distinction between the intracellular and the extracellular proteostasis should be highlighted.As mentioned above, the intracellular network can “repair” misfolded intracellular proteins by the ATP-driven chaperones, whereas in the current thinking the ECs mostly hold misfolded extracellular proteins for subsequent destruction.
In the revised manuscript (section 2.2), we have added new text to explicitly flag the fundamental differences between intra- and extra-cellular proteostasis.
- The authors may want to list specific questions regarding the ECs in an expanded Conclusion section to inspire new researchers in this area.
To the closing sentences of the Conclusion, in the revised manuscript we have added several unanswered questions to stimulate future research.
- Line 31: “some chaperones are constitutively secreted into the extracellular milieu.” This needs references.
An appropriate reference has been cited at this point in the revised manuscript.
- Line 106-107: “Extracellular misfolded proteins and small soluble aggregates are thought to be bound by ECs and cleared from the body by receptor-mediated endocytosis and subsequent lysosomal degradation.” This needs references.
Citations of several relevant references have been been added at this point in the revised manuscript.
- Line 116-117: “ECs and the PAS may co-operatively work together as key elements of extracellular proteostasis to ensure the timely and safe removal of misfolded and aggregating proteins from body fluids.” Is this purely speculative? Otherwise this needs references.
Apologies for not making this clear. This statement is a hypothesis supported by published experimental work. In the revised manuscript, this sentence has been modified to read: "It has been shown that two ECs (clusterin and a2-macroglobulin) bind to toxic protein fragments released by the plasmin-mediated digestion of amorphous protein aggregates, leading to the proposal that ECs and the PAS may work co-operatively to ensure the timely and safe removal of misfolded and aggregating proteins from body fluids." and the corresponding supporting reference is now cited following this sentence.
Reviewer 2 Report
This is a timely and interesting review, dealing with extracellular chaperones. It is my opinion that this work should be published in the present format.
Author Response
Reviewer 2
This is a timely and interesting review, dealing with extracellular chaperones. It is my opinion that this work should be published in the present format.
We thank the reviewer for their comments.
Reviewer 3 Report
This is an interesting and informative read. I have three minor suggestions:
1. There is not much by the way of highlighted unanswered questions, especially in the conclusions. That would make this review more helpful.
2. One specific question I have is has anyone yet evaluated the role of LLPS or similar phase diagrams with these systems?
3. I am curious about whether the model systems are all of equivalent value in the studies cited.
Author Response
Reviewer 3
This is an interesting and informative read. I have three minor suggestions:
- There is not much by the way of highlighted unanswered questions, especially in the conclusions. That would make this review more helpful.
To the closing sentences of the Conclusion, in the revised manuscript we have added several unanswered questions to stimulate future research.
- One specific question I have is has anyone yet evaluated the role of LLPS or similar phase diagrams with these systems?
The interactions of ECs with liquid-liquid phase separation (LLPS) has yet to be investigated. As there is no data available addressing this, we have made no comment on this in the revised manuscript.
- I am curious about whether the model systems are all of equivalent value in the studies cited.
We apologise, but it is unclear from this comment which "model systems" are being referred to. We are happy to address this point if further clarification can be provided.